# The Importance of Increased Serum GFAP and UCH-L1 Levels in Distinguishing Large Vessel from Small Vessel Occlusion in Acute Ischemic Stroke

**DOI:** 10.3390/biomedicines12030608

**Published:** 2024-03-07

**Authors:** Ivan Kraljević, Sara Sablić, Maja Marinović Guić, Danijela Budimir Mršić, Ivana Štula, Krešimir Dolić, Benjamin Benzon, Vana Košta, Krešimir Čaljkušić, Marino Marčić, Daniela Šupe Domić, Sanja Lovrić Kojundžić

**Affiliations:** 1Clinical Department of Diagnostic and Interventional Radiology, University Hospital of Split, 21000 Split, Croatiamaja.marinovic.guic@gmail.com (M.M.G.); danijelabudimir@gmail.com (D.B.M.); ivanast23@yahoo.com (I.Š.); kdolic79@gmail.com (K.D.); 2School of Medicine, University of Split, 21000 Split, Croatia; benzon.benjamin@gmail.com (B.B.); vanakosta@gmail.com (V.K.); kresimir.caljkusic@gmail.com (K.Č.); marino.marcic@yahoo.com (M.M.); 3University Department of Health Studies of the University of Split, 21000 Split, Croatia; daniela.supedomic@gmail.com; 4Department of Neurology, University Hospital of Split, 21000 Split, Croatia; 5Department of Medical Laboratory Diagnostics, University Hospital of Split, 21000 Split, Croatia

**Keywords:** ischemic stroke, biomarker, UCH-L1, GFAP, large vessel occlusion, small vessel occlusion

## Abstract

Acute ischemic stroke (AIS) is one of the leading causes of morbidity worldwide, thus, early recognition is essential to accelerate treatment. The only definite way to diagnose AIS is radiological imaging, which is limited to hospitals. However, two serum neuromarkers, glial fibrillary acidic protein (GFAP) and ubiquitin C-terminal hydrolase-L1 (UCH-L1), have been proven as indicators of brain trauma and AIS. We aimed to investigate the potential utility of these markers in distinguishing between large vessel occlusion (LVO) and small vessel occlusion (SVO), considering differences in treatment. Sixty-nine AIS patients were included in our study and divided into LVO and SVO groups based on radiological imaging. Control group consisted of 22 participants without history of neurological disorders. Results showed differences in serum levels of both GFAP and UHC-L1 between all groups; control vs. SVO vs. LVO (GFAP: 30.19 pg/mL vs. 58.6 pg/mL vs. 321.3 pg/mL; UCH-L1: 117.7 pg/mL vs. 251.8 pg/mL vs. 573.1 pg/mL; *p* < 0.0001), with LVO having the highest values. Other prognostic factors of stroke severity were analyzed and did not correlate with serum biomarkers. In conclusion, a combination of GFAP and UCH-L1 could potentially be a valuable diagnostic tool for differentiating LVO and SVO in AIS patients.

## 1. Introduction

Acute ischemic stroke (AIS) continues to rank among the leading causes of morbidity and disability in the world [1]. It is important to establish a quick and accurate diagnosis in patients with suspected AIS and definitively distinguish it from intracerebral haemorrhage (ICH) and stroke mimics in order to accelerate triage and ultimately enhance patient outcomes [2,3]. Generally, the detection of stroke initially relies on clinical presentation and urgent neuroimaging performed by computed tomography (CT) or magnetic resonance imaging (MRI). While unenhanced brain CT can easily identify cases of ICH, it is relatively insensitive at identifying acute and small ischemic strokes. On the other hand, certain MRI sequences, like DWI (diffusion-weighted imaging), are more sensitive at detecting early signs of stroke, even within 30 min of stroke onset. Unfortunately, only specialized hospitals with on-call neuroradiologists have access to this expensive equipment and technology, and imaging is reliant on the stillness of the patient for a longer time period than during CT imaging [4,5]. Therefore, better pre-hospital stroke identification tools are necessary.

Highly specific serum biomarkers released during AIS were suggested to be valuable diagnostic tools in addition to current routine diagnostic methods. This is similar to other diseases, such as myocardial infarction, in which blood biomarkers have been widely used in clinical management [6]. They could provide an objective and rapid tool for the early diagnosis, triage, and prognosis of stroke patients. Recently, a combined biomarker test including glial fibrillary acidic protein (GFAP) and ubiquitin C-terminal hydrolase-L1 (UCH-L1) has been used in patients with mild traumatic brain injury (TBI) before neuroimaging procedures [7]. GFAP is a protein that is primarily expressed in astrocytes; therefore, it is not released under physiological conditions, and its serum levels in healthy people are very low [8,9]. The enzyme UCH-L1 is essential for the self-repair processes of neurons and neuroendocrine cells, where it is present in large quantities [10]. Over the past few years, extensive research has been conducted on the usefulness of GFAP in acute stroke, whereas in the case of UCH-L1, there is much less data. Several multicenter studies showed a slow release of GFAP in AIS compared to the rapid increase in ICH patients [11,12,13,14]. An animal model study demonstrated higher levels of UCH-L1 in rats subjected to AIS compared to rats with ICH [15]. Another study tested a combination of UCH-L1 and GFAP in stroke patients and showed no usefulness of UCH-L1 regarding the differentiation of AIS and ICH [16].

A better, timely diagnostic is imperative for potentially successful treatment. Over the last decade, treatment of AIS has greatly evolved and includes mechanical thrombectomy (MT) for large vessel occlusions (LVO) in addition to single intravenous thrombolysis (IVT) used for any type of AIS within the limited window of 4.5 h [17,18]. There is no standardized definition for LVO, but it commonly includes the intracranial internal carotid artery (ICA), proximal segments of the middle cerebral artery (M1 and M2), proximal anterior cerebral artery (A1), or basilar artery (BA), as well as combinations of these vessels [19]. MT is usually performed at a comprehensive stroke center (CSC), a hospital with complex endovascular facilities. Patients sent directly to a CSC for MT adhere to the “mothership” model. The “drip and ship” pathway, also referred to as the primary care pathway, is an alternate route in which patients are referred to the closest stroke center that offers IVT as soon as possible [20]. In contrast, small vessel occlusions (SVO) include distal segments of all aforementioned vessels and have one of the traditional clinical lacunar syndromes, including pure motor stroke, pure sensorimotor stroke, pure sensory stroke, ataxic hemiparesis, or clumsy hand dysarthria. These patients are not eligible for MT [21]. Considering the above-mentioned differences in treatment and the need for emergent neuroimaging, an easily assessed biomarker would be of great help for the first-in-line distinction between LVO and SVO.

The aim of this study was to explore the diagnostic value of a combined GFAP and UCH-L1 biomarker test in the context of acute ischemic stroke (both LVO and SVO) and the possibility of differentiating between them based on serum levels. We were particularly interested in evaluating the markers in the early time window when patients first presented to the emergency department.

## 2. Materials and Methods

### 2.1. Study Design and Participants

Patients who presented with symptoms of AIS (sudden occurrence of a focal neurological deficit) to the Neurology Department of the University Hospital of Split, Croatia, underwent standard medical evaluation and assessment using the National Institutes of Health Stroke Scale (NIHSS) [22]. Their blood samples were also collected. If the symptoms started within the last 6 h, a prompt neuroimaging protocol for AIS was requested. The standard neuroimaging protocol for AIS in our hospital includes unenhanced brain CT, CT angiography (CTA), and CT perfusion (CTP) imaging. If there were signs of ICH, acute hematomas, tumors, or demarcated hypodense lesions on unenhanced CT, further CTA and CTP were not obtained. Based on clinical and imaging data, inclusion criteria were admission within 6 h of symptom onset, presence of neurological symptoms at the time of admission, and radiologically verified AIS; LVO or SVO. Exclusion criteria were previous strokes, head trauma, ICH, intracranial tumors, and severe brain edema. In the period from March 2023 to July 2023, sixty-nine patients met our inclusion criteria. Controls consisted of 22 volunteers who did not have any focal neurological deficits or antecedents of central nervous system disease. They were not age and sex matched to AIS patients considering differences between LVO and SVO groups, but they had a similar comorbidity burden. Patient demographics (age, sex, risk factors, comorbidities, and previous medication history), clinical (NIHSS on admission) and radiological data (stroke location, positive “hyperdense sign”—focal hyperdensity on an unenhanced CT suggesting a clot), and general laboratory findings were collected from the hospital information system.

### 2.2. Blood Sample Collection and Processing

Blood samples were taken as part of the routine workup upon hospital admission, with additional samples for UCH-L1 and GFAP. Within 60 min of blood collection, blood tubes were transported to the hospital’s laboratory and centrifuged at 1500× *g* for 10 min. After that, the serum was divided into 0.5 mL aliquots and kept in storage at −80 °C. Blood samples were recovered from the storage after determining which patient had AIS, and after distribution into SVO and LVO groups. Blood samples of patients that did not meet our inclusion criteria were not analyzed. Board-certified laboratory technicians, blinded to clinical information, handled serum samples. The commercial Abbott kit (04W1720), Abbott, Sligo, Ireland was utilized to measure the levels of UCH-L1 and GFAP in the serum. Every measurement was performed in complete calibration mode. Ranges of measurable values within which the results can be reported are based on representative data for the limit of quantification (LoQ) and limit of detection (LoD) and are, for GFAP 6.1–42,000 pg/mL, for UCH-L1 26.3–25,000 pg/mL, which is in accordance with the definitions of CLSI guidelines EP34, 1. edition [23].

### 2.3. Imaging Data and Analysis

Using a 128-slice CT Siemens Somatom, Erlangen, Germany, all patients underwent a standardized neuroimaging protocol for AIS, analyzed using the Syngovia software (version VB60A, Hofix05), Siemens Healthineers, Erlangen, Germany. Two experienced radiologists, including an interventional radiologist, analyzed unenhanced brain CT, CTA, and CTP for signs of AIS (hypodense demarcated area on unenhanced CT, positive “hyperdense sign”, vessel occlusion, lowered brain perfusion with a core/penumbra mismatch ratio denoting salvageable brain tissue in the penumbra area), and if there were indications for MT. Based on their findings, patients were divided into SVO and LVO groups. We defined LVO as an occlusion of ICA, M1, M2, A1, BA, or any combination of those vessels, such as T-occlusion (distal ICA, M1, and A1) or tandem occlusion (distal ICA and M1). Positive “hyperdense sign” was analyzed in both the anterior and posterior brain circulation, while the occlusion of multiple vessels was analyzed only in the anterior circulation, due to the difficulty of assessing the propagation of the thrombus from BA into its smaller, anatomically variant branches. SVO was defined as an occlusion in distal segments of anterior, middle, or posterior cerebral arteries, as well as cerebellar arteries. Patients in the LVO group underwent MT at our CSC.

### 2.4. Statistical Analysis

Continuous data are presented as geometric mean and geometric standard deviation factor, if not stated otherwise, since most of the data were distributed log normally. Discrete data are presented as fractions (%). Since groups in this study were ordered categories, linear test for trend was used as a mainstay of analysis for continuous variables furthermore, unpaired *t* test to which the former reduces in case of comparisons between two groups was also used. If data were log normally distributed, they were log transformed before testing procedure. Differences in fractions were tested by Chi square test. As a statistical measure of evidence *p* values and coefficient of determination (η^2^ or R^2^) were used. To study diagnostic characteristics of the biomarkers, logistic regression modelling was used. Classification threshold (cut-off) was selected to the one that maximizes accuracy. Along with the standard classification metrics, Tjur’s R^2^ and *p* value from likelihood ratio test (H_0_ intercept only model) are given.

## 3. Results

This study included a total of 91 participants: 69 patients with AIS and 22 controls. AIS patients were further divided into SVO group with 29 patients, and LVO group with 40 patients. Their baseline characteristics were shown in Table 1 with general laboratory findings for AIS patients in Table 2.

Serum concentrations of GFAP and UCH-L1 were different between all study groups. The control group showed the lowest serum levels (geometric mean: 30.19 pg/mL for GFAP, 117.7 pg/mL for UCH-L1), SVO somewhat higher serum levels (geometric mean: 58.6 pg/mL for GFAP, 251.8 pg/mL for UCH-L1), with highest values in LVO group (geometric mean: 321.3 pg/mL for GFAP, 573.1 pg/mL for UCH-L1), showing a positive linear trend, *p* < 0.0001 (Table 3 and Figure 1).

The diagnostic accuracy of GFAP as a biomarker showed a 77% accuracy in discriminating between LVO and SVO, with a cut-off value of 200.53 pg/mL. On the other hand, discrimination between LVO and the control group demonstrated 89% accuracy, with a cut-off value of 63.74 pg/mL. Further, UCH-L1 as a biomarker was accurate in approximately 70% of cases when it comes to LVO vs. SVO discrimination, with a cut-off value of 498.89 pg/mL, and an accuracy of 95% in discriminating between LVO and the control group, with a cut-off value of 288.09 pg/mL (Table 4).

We investigated if there was a correlation between biomarker serum levels and NIHSS scores in patients with LVO. Patients were divided into 2 groups: those with favorable NIHSS (0–15), and those with unfavorable NIHSS (16–42). Both GFAP and UCH-L1 values did not correlate with NIHSS severity. We also assessed if the presence of a positive “hyperdense sign” in patients with LVO showed a difference in GFAP and UCH-L1 levels compared to a negative “hyperdense sign”. Serum values of GFAP and UCH-L1 showed no comparable differences between the negative and positive groups. Another radiological finding that we compared in patients with LVO was the location of the thrombus in the anterior circulation, whether it affected a single vessel or multiple (combination of occluded vessels—T-occlusion or tandem occlusion). Posterior circulation occlusions were excluded only for this variable. Again, both GFAP and UCH-L1 showed no differences in serum levels between the single vessel and multiple vessel groups (Table 5).

## 4. Discussion

Our study showed GFAP and UCH-L1 at significantly elevated levels in AIS patients compared to neurologically healthy controls, as well as a difference in their levels between SVO and LVO patients. Other prognostic factors, such as NIHSS score, positive “hyperdense sign”, and location of the clot, did not correlate with serum biomarkers in LVO patients.

Considering that UCH-L1 is highly and specifically expressed in neurons and that GFAP is almost exclusively produced in astrocytes and released when the astrocyte cytoskeleton disintegrates, it would be expected that the destruction of these cells would produce detectable levels of biomarkers in the serum [24,25]. This is consistent with studies that tested the aforementioned biomarkers in a setting of acute neuronal damage or neuronal death [16,26,27,28,29]. Diaz-Arrastia et al. showed that GFAP and UCH-L1 both separately and in combination had strong sensitivity for differentiating between TBI patients and healthy controls [26]. Yigit et al. demonstrated that UCH-L1 levels were significantly higher in patients with AIS and ICH compared to healthy controls but did not differ between the AIS and ICH groups [27]. On the other hand, Ren et al. showed that serum levels of GFAP can be a possible tool for early distinction of AIS and ICH [16]. Furthermore, Wu et al. illustrated that both GFAP and UCH-L1 were partially elevated in sepsis-associated encephalopathy and were linked to a poor prognosis and a lower quality of life [29]. A common finding in all mentioned studies is neuronal injury. Although they originate in different cells, GFAP and UCH-L1 could be used complementarily as they reflect different injury mechanisms. Therefore, it would be expected that the greater the volume of destroyed brain tissue, the higher the detectable levels of UCH-L1 and GFAP.

On that note, Papa et al. showed that UCH-L1 and GFAP measured in human serum after mild and moderate TBI can be found within an hour of injury and are linked to injury severity indicators such as Glasgow coma score and CT lesions [30,31]. These studies illustrated that a smaller volume of neuronal injury can also elevate serum biomarkers, which is in accordance with our study showing that even SVO can be distinguished from a healthy brain based on GFAP and UCH-L1 levels.

Although most studies on GFAP serum levels have sought to determine differences between AIS and ICH, some studies have focused solely on time-dependent changes of GFAP in AIS [32]. Aurell et al. determined that the temporary release of astroglial proteins, including GFAP, into the cerebrospinal fluid (CSF) may indicate localized ischemic injury and, in subsequent stages, the degeneration of astroglial cells in the penumbra zone [33]. In addition, Herrmann et al. analyzed post-ischemic release patterns of GFAP and protein S-100B in serum, which showed correlation with clinical and morphological outcomes of AIS [34]. On the other hand, UCH-L1 has limited data in clinical trials on AIS patients in determining the severity of neurological deficit. An animal study by Liu et al. showed elevated levels of UCH-L1 after 30 min and 2 h in models of ischemic strokes in rats, measured in serum and CSF, but similar findings have not yet been confirmed in humans [35].

Only one study by Hu et al. compared both GFAP and UCH-L1 serum levels in AIS patients with healthy controls and determined increased levels at the early stage of stroke with a certain correlation to the severity of cerebral infarction, which is consistent with our results [36]. However, although they divided the severity of AIS into a heavy, medium, and light groups, the division only corresponds to the occlusion outcome, that is, the volume of affected brain tissue, disregarding differentiation of the underlying cause—vessel occlusion. The main importance of our study is differentiating two types of occlusion as a cause of AIS in correlation with these biomarkers, which ultimately have different therapeutic approaches. Rapid determination of the level of occlusion and hastening possible endovascular treatment (EVT) for patients with LVO, especially for those living far from a CSC, can be lifesaving. An RCT, as a part of a multicenter randomized clinical trial, MR CLEAN, showed that the initial benefit of EVT declines with each hour of reperfusion delay, and the absolute risk difference for a favorable result drops by 6% with each hour of delay [37]. Hence, any possible way of hastening the diagnosis of AIS, primarily in the pre-hospital setting, increases survivability and decreases consequential morbidity.

We also compared general laboratory results between the SVO and LVO groups. Only PT was significantly different, being somewhat lower in the LVO group. By only analyzing PT and not any of the coagulation factors, which are not routinely measured, it is difficult to draw any conclusions. Numerous biomarkers related to coagulation and inflammation have been linked to AIS, but they are still unable to predict the clinical result of treatment, the severity of stroke in the acute phase, or patients who are at risk [38]. Also, none of the general markers of tissue damage, such as LDH and CRP, showed differences between the examined groups [39,40]. Although the number of damaged cells in LVO is much greater than SVO, serum levels of these markers are not disparate. On the other hand, serum levels of GFAP and UCH-L1 correlate to the size of damaged tissue, which could imply their specificity in registering brain tissue damage.

Furthermore, we investigated the correlation of stroke severity parameters and predictive factors—NIHSS score, positive “hyperdense sign”, and the combination of affected vessels, with GFAP and UCH-L1 serum levels in LVO patients [22,41]. Some studies showed a positive correlation between GFAP serum levels and NIHSS score. The higher the value of GFAP serum level, the higher the value for NIHSS [42,43]. That is not in accordance with our result, which showed no difference between favorable and unfavorable NIHSS groups. However, these studies did not have a consistent way of dividing groups by NIHSS score and used different time periods in which they first sampled patients’ blood, including up to 72 h from symptom onset. On the other hand, a study by Yigit et al. showed no correlation between UCH-L1 and NIHSS score, which is comparable to our results [27]. Our division for NIHSS groups followed categorization for mild, moderate, moderate to severe, and severe strokes, with mild and moderate considered favorable, and moderate to severe and severe considered unfavorable. However, this grouping is based entirely on clinical performance of the patient at symptom onset, when neuronal damage may not have happened as yet. Hypoperfusion of salvageable brain tissue in the penumbra zone could cause severe symptoms without raising biomarker serum levels at the moment of our testing.

Another parameter suggesting a more serious clinical outcome (in terms of more frequent impairment of vision, motor function of upper and lower extremities, dysarthria, or dysphasia) is a positive “hyperdense sign” [41]. Our study was the first to investigate the correlation of this predictive variable with GFAP or UCH-L1 serum levels, and it showed no difference between the studied groups. One possible explanation is that a positive “hyperdense sign” only suggests a larger or harder thrombus, more difficult for IVT or MT treatment. Subsequent, more serious clinical outcomes would then depend on the success of said treatment, not the size of the affected brain. Once a vessel is occluded, the characteristics of the analyzed thrombus are irrelevant for the hypoperfused neurons distal to the occlusion site. Therefore, levels of GFAP and UCH-L1 would not depend on the density of the thrombus, as suggested by hyperattenuation on CT. Similarly, whether one large vessel or a combination of them were occluded seemed to have no effect on the GFAP and UCH-L1 serum levels. This could also be explained by a similar mechanism. The described combinations of occluded vessels–T-occlusion and tandem occlusion–tend to start at the most proximal part and propagate distally [44]. Thus, after the initial occlusion and distal ischemia have occurred, further occlusions in the distal branches do not contribute to additional brain tissue damage and should have no supplemental effect on the GFAP and UCH-L1 serum levels.

One limitation of our study is that there was an age difference between all participant groups. However, contradictory results are reported in the literature on GFAP and UCH-L1 serum levels increasing with age, where some show no difference between age groups and some show increasing levels with age [27,45]. Also, there was a difference in sex distribution between our studied groups. Predominantly, women presented with LVO, while mostly men presented with SVO. A recent study displayed higher concentrations of UCH-L1 in male patients compared to female, while similar findings were not found with GFAP. Regardless of these findings, the overall diagnostic precision did not significantly differ between sex groups [46]. Although there are differences in age and sex between our groups, their distribution reflects the general population and epidemiology of stroke, with older age and female sex linked to a worse burden of stroke mortality and disability [47]. Finally, we included a relatively small number of patients. A larger study sample could provide further insight into the potential use of GFAP and UCH-L1 in AIS patients by correlating them with functional recovery after different outcomes of MT or IVT and follow-ups after 3 months. Division into smaller groups based on treatment results within our study sample would be inefficient, as those groups would be too small for statistical analysis. That will be a task for future studies on GFAP and UCH-L1 in AIS.

The main advantage of this study is proving that there were differences in GFAP and UCH-L1 serum levels between SVO and LVO patients and, also, furthering knowledge of less-researched UCH-L1 in the setting of AIS. The possibility of determining a cut-off biomarker value for LVO with relatively high sensitivity and specificity shows promising potential for future research. It could eventually help develop rapid biomarker tests that could be a complementary tool in the early, pre-hospital detection of AIS, similar to the role of rapid troponin test for myocardial infarction [6]. In countries where a smaller percentage of the population lives in cities with a CSC, others rely on early symptom recognition by themselves, by paramedics or by emergency medicine doctors, on adequate diagnosis, first-in-line radiological imaging (if available) that is then sent to a CSC via teleradiology, and later, on functional transportation system. During this lengthy process, precious time is lost. Therefore, we hope further studies on a larger population sample would provide more stable conclusions regarding precise guidelines about bloodwork done before, or in some cases, instead of imaging.

## 5. Conclusions

Serum levels of biomarkers GFAP and UCH-L1 are distinguishable between LVO and SVO in AIS patients, and neurologically healthy controls. LVO patients showed the highest biomarker values, and controls showed the lowest values. The determined threshold values for both biomarkers have fairly good sensitivity and specificity at distinguishing LVO from SVO, and excellent sensitivity and specificity when discriminating LVO and controls. Therefore, further research could possibly prove them to be an effective complementary tool in the early, pre-hospital detection of AIS, and early determination of patients eligible for EVT. However, other prognostic factors, such as NIHSS score, positive “hyperdense sign”, and location of the clot, did not correlate with serum biomarkers in LVO patients.

## Figures and Tables

**Figure 1 biomedicines-12-00608-f001:**
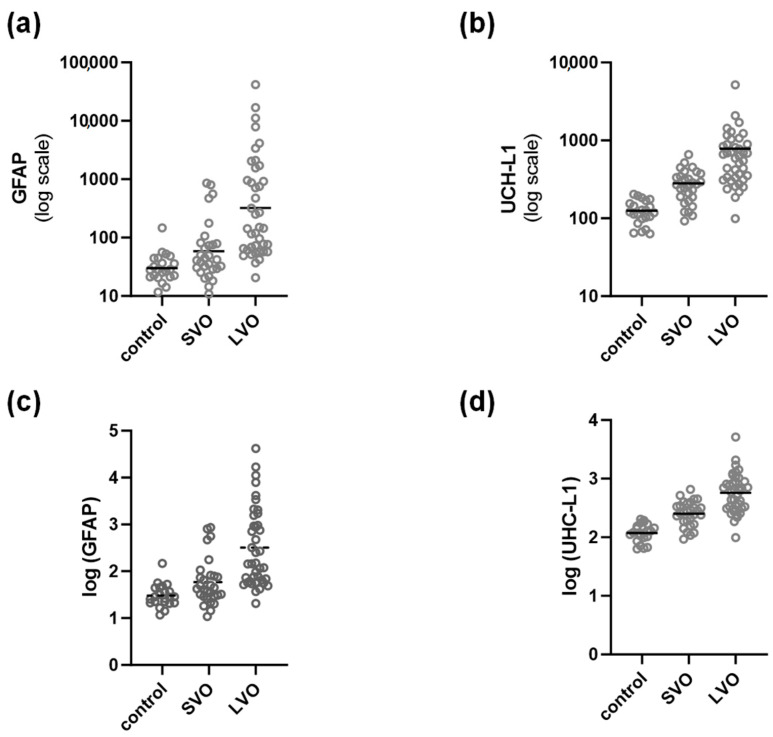
Relationship between GFAP, UHCL-1 and size of occluded vessel. (**a**,**b**) Data are not log transformed, however they are depicted on log scale, averages are presented as geometric means. (**c**,**d**) Data are log transformed and averages are presented as arithmetic average. Control, patients without blood vessel occlusions; SVO = small vessel occlusion; LVO = large vessel occlusion; GFAP = glial fibrillary acidic protein; UCH-L1 = ubiquitin C-terminal hydrolase-L1.

**Table 1 biomedicines-12-00608-t001:** Baseline characteristics of the study population. Data are given as geometric mean with geometric standard deviation factor or as number and percentage.

	Control (*n* = 22)	SVO (*n* = 29)	LVO (*n* = 40)	*p* Value
Age, years (GM, GSDF)	66.77, 1.09	72.03, 1.16	77.6, 1.15	<0.0001 *
Women (*n*, %)	10 (45.45%)	9 (31.03%)	29 (72.5%)	0.002 ^#^
Arterial hypertension (*n*, %)	18 (81.82%)	21 (72.41%)	29 (72.5%)	0.722 ^#^
Diabetes (*n*, %)	5 (22.73%)	5 (17.24%)	7 (17.5%)	0.867 ^#^
Atrial fibrillation (*n*, %)	9 (40.91%)	10 (34.48%)	15 (37.5%)	0.89 ^#^

* Test for linear trend; ^#^ Chi square test; SVO = small vessel occlusion; LVO = large vessel occlusion; GM = geometric mean; GSDF = geometric standard deviation factor.

**Table 2 biomedicines-12-00608-t002:** General laboratory findings in different AIS groups (SVO and LVO). Data are given as geometric mean with geometric standard deviation factor.

	SVO (*n* = 29)		LVO (*n* = 40)		*p* Value *	R^2^ Value *
	GM	GSDF	GM	GSDF		
Leukocytes (number × 10^9^)	8.243	1.42	8.944	1.387	0.332	1.45%
Neutrophils (%)	67.53	1.193	72.54	1.197	0.079	4.68%
Lymphocytes (%)	18.39	1.834	15.61	1.665	0.154	3.1%
Na^+^ (mmol/L)	140.7	1.02	140.8	1.028	0.852	0.05%
K^+^ (mmol/L)	4.117	1.102	3.94	1.14	0.142	3.4%
LDH (mmol/L)	178.8	1.66	211.8	1.292	0.212	3.16%
CRP (mg/L)	3.335	3.507	5.347	3.914	0.154	3.1%
PT (s)	1.141	1.178	0.927	1.408	0.0019	15.13%

* Unpaired *t* test; SVO = small vessel occlusion; LVO = large vessel occlusion; GM = geometric mean; GSDF = geometric standard deviation factor; Na^+^ = sodium; K^+^ = potassium; LDH = lactate dehydrogenase; CRP = C-reactive protein; PT = prothrombin time.

**Table 3 biomedicines-12-00608-t003:** GFAP and UCH-L1 serum levels in AIS patients (SVO and LVO) and controls. Data are given as geometric mean with geometric standard deviation factor.

	Control (*n* = 22)		SVO (*n* = 29)		LVO (*n* = 40)		*p* Value *	R^2^ Value *
	GM	GSDF	GM	GSDF	GM	GSDF		
GFAP (pg/mL)	30.19	1.72	58.6	3.172	321.3	6.93	*p* < 0.0001	31.32%
UCH-L1 (pg/mL)	117.7	1.429	251.8	1.637	573.1	2.129		53.74%

* Test for linear trend; SVO = small vessel occlusion; LVO = large vessel occlusion; GM = geometric mean; GSDF = geometric standard deviation factor; GFAP = glial fibrillary acidic protein; UCH-L1 = ubiquitin C-terminal hydrolase-L1.

**Table 4 biomedicines-12-00608-t004:** Diagnostic characteristics of GFAP and UCH-L1.

	GFAP		UCH-L1	
Outcome	LVO vs. SVO *	LVO vs. Control ^#^	LVO vs. SVO ^†^	LVO vs. Control ^‡^
AUC	0.7996	0.9403	0.8159	0.9795
specificity (%)	62.07	81.82	62.07	95.45
sensitivity (%)	85.5	92.5	75	95
accuracy (%)	76.81	88.71	69.57	95.16
cut-off (pg/mL)	200.53	63.74	498.89	288.09

* Equation for predicting log odds for LVO vs. SVO: logit (LVO) = 1.775∙log(γ(GFAP)) − 3.332; 0.43 (43%) was used as probability threshold (cut-off) for classification, Tjur’s R^2^ = 0.2273, *p* < 0.0001; ^#^ Equation for predicting log odds for LVO vs. control: logit (LVO) = 7.007∙log(γ(GFAP)) − 11.77; 0.5 (50%) was used as probability threshold (cut-off) for classification, Tjur’s R^2^ = 0.577, *p* < 0.0001; ^†^ Equation for predicting log odds for LVO vs. SVO: logit (LVO) = 5.063∙log(γ(UCH-L1)) − 12.66; 0.5 (50%) was used as probability threshold (cut-off) for classification, Tjur’s R^2^ = 0.3033, *p* < 0.0001; ^‡^ Equation for predicting log odds for LVO vs. control: logit(LVO) = 14.8∙log(γ(UCH-L1)) − 33.9; 0.5 (50%) was used as probability threshold (cut-off) for classification, Tjur’s R^2^ = 0.8023, *p* < 0.0001; GFAP = glial fibrillary acidic protein; UCH-L1 = ubiquitin C-terminal hydrolase-L1; AUC = area under the curve.

**Table 5 biomedicines-12-00608-t005:** Correlation of stroke severity prediction factors (NIHSS, “hyperdense sign” and multiple vessel occlusion) with GFAP and UCH-L1 serum levels LVO patients. Data are given as geometric mean with geometric standard deviation factor.

		GFAP (pg/mL)	UCH-L1 (pg/mL)
Favorable NIHSS (*n* = 16)	GM	300.9	497.8
	GSDF	7.82	2.51
Unfavorable NIHSS (*n* = 13)	GM	298.2	565.7
	GSDF	4.595	1.997
*p* Value *		0.395	0.682
R^2^ Value *		2.7%	0.63%
Positive “hyperdense sign” (*n* = 23)	GM	286.8	525.4
	GSDF	4.9	1.73
Negative “hyperdense sign” (*n* = 17)	GM	480.4	644.6
	GSDF	10	2.655
*p* Value *		0.934	0.405
R^2^ Value *		0.02%	1.8%
Multiple vessel occlusion (*n* = 13)	GM	242.6	524.3
	GSDF	8.81	2.21
Single vessel occlusion (*n* = 25)	GM	369.6	608.4
	GSDF	6.758	2.1
*p* Value *		0.526	0.57
R^2^ Value *		1.1%	0.9%

* Test for linear trend; NIHSS = with National Institutes of Health Stroke Scale; LVO = large vessel occlusion; GM = geometric mean; GSDF = geometric standard deviation factor; GFAP = glial fibrillary acidic protein; UCH-L1 = ubiquitin C-terminal hydrolase-L1. Note: Some patients in the LVO group did not have a complete medical history; therefore, their NIHSS scores were N/A for statistical analysis.

## Data Availability

Data are contained within the article. The datasets are available upon reasonable request to the corresponding author.

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
