# Peer review of "The Importance of Increased Serum GFAP and UCH-L1 Levels in Distinguishing Large Vessel from Small Vessel Occlusion in Acute Ischemic Stroke"

_biomedicines, 2024, doi:10.3390/biomedicines12030608_

Round 1
Reviewer 1 Report
Comments and Suggestions for Authors
The distinction between small and large vessel occlusion biochemical data is interesting, but there is a large overlap: though both patient data are higher than asymptomatic volunteers. After all evaluations, however, it is unclear how a cut-off value of biochemical markers might improve the rapid detection of the size of the occluded vessel. Unclear conclusions.
Author Response
We extend our gratitude for investing your valuable time in thoroughly evaluating our manuscript and for raising adequate concerns regarding our research. There are some limitations in our study that we tried to further explain in our revision.
Our study groups (controls, SVO and LVO) were not age and sex matched. Since we noticed that our examined (SVO and LVO) groups differ in these parameters we could not match our control group to both examined groups. Furthermore, some studies showed no impact of these variables on examined data as explained in the discussion. On top of that, all other parameters regarding comorbidities that could affect outcomes were matched. We added this explanation in the “2.1 subsection of the Materials and Methods”.
Also we added a few supplemental clarifications on study design and methods in the “2.1 and 2.2 subsections of the Materials and Methods”.
The main goal of our research was to determine if there are differences between examined groups. We have also now added threshold values for both GFAP and UCH-L1 biomarkers, along with accuracy, specificity and sensitivity of both tests for determining values of LVO in contrast to SVO and control groups, as only patients in LVO group can benefit from emergent endovascular treatment. According to that, we modified the text in “section 3 Results”, as well as added statistical explanation in the “2.4 subsection of Material and Methods”.
We have also revised our conclusions (in “sections 4 Discussion, and 5 Conclusion”) and given the context in which these results could be applied in the future; they could help in eventual development of rapid biomarker tests that could be used for prehospital triage, similar to rapid troponin tests for myocardial infarction.
We hope our revisions and explanations are acceptable.
Reviewer 2 Report
Comments and Suggestions for Authors
The authors present an interesting study on blood tests in AIS patients potentially aiding in the pre-selection process of said patients and ultimately helping to decide for the correct treatment.
The study is well conducted, the statistics appropriate and the quality of the writing sufficient.
I have only few points that could be adressed:
1. Is there a correlation (even if not statistically significant) between the measured blood values and the severity of the outcome after therapy?
2. Can you point to a certain threshold value where e.g. 95% of patients are likely to have LVO?
3. Please add some practical considerations to the findings, e.g. would you recommend every patient to have bloodwork done before imaging? Or in what scenarios would you do the testing?
Author Response
We are thankful for a great review. We are happy to have received these well-thought questions and are thankful for the opportunity to address them.
Comments 1: Is there a correlation (even if not statistically significant) between the measured blood values and the severity of the outcome after therapy?
Response 1: We did not correlate values of biomarkers with outcome severity since it would be necessary to further subdivide a relative small number of patients into several, even smaller groups based on outcomes of mechanical thrombectomy and/or intravenous thrombolysis. Those groups would be inadequate for relevant statistical analysis. Furthermore, there are several factors that play a part in clinical outcomes – comorbidities such as heart failure, pneumonia, or urosepsis that can be a precipitating cause of AIS, or a consequence of hospitalization. All of that cannot be taken into account when assessing only biomarker levels.
Comments 2: Can you point to a certain threshold value where e.g. 95% of patients are likely to have LVO?
Response 2: The main goal of our research was to determine if there are differences between examined groups. We have also now added threshold values for both GFAP and UCH-L1 biomarkers, along with accuracy, specificity and sensitivity of both tests for determining values of LVO in contrast to SVO and control groups, as only patients in LVO group can benefit from emergent endovascular treatment. Thus, we modified the text in "subsection 2.4 of Materials and Methods" and “section 3 Results”. Also, some of our conclusions were revised accordingly in “section 5 Conclusion”.
Comments 3: Please add some practical considerations to the findings, e.g. would you recommend every patient to have bloodwork done before imaging? Or in what scenarios would you do the testing?
Response 3: Thank you for the excellent proposal. Our findings, on their own, cannot be used to recommend bloodwork for every patient with AIS, as this is the first study on a relatively small sample. However, further investigations on the usefulness of GFAP and UCH-L1 in SVO and LVO AIS can help in development of rapid biomarker tests that could be used for prehospital triage, similar to rapid troponin tests for myocardial infarction. In remote places without radiological diagnostics, or on-call radiologists, or even in hospitals that are not comprehensive stroke centers, rapid tests would be very useful in determining the need for quick patient transportation. In our country, only 28.5% of the population lives in cities with a comprehensive stroke center that can provide quick treatment. Other 71.5% of the population relies on early symptom recognition by themselves, paramedics or emergency medicine doctors, adequate diagnosis, first-in-line radiological imaging (if available) that is then sent to a comprehensive stroke center via teleradiology, and later, on functional transportation system. During this lengthy process, precious time is lost. Therefore, we hope further studies on larger population sample would definitely provide more stable conclusions regarding precise guidelines about bloodwork done before or in some cases instead of imaging. The same is mentioned in the manuscript, within the last section of Discussion.
Round 2
Reviewer 1 Report
Comments and Suggestions for Authors
Well corrected: accept now